# Male Infertility: Shining a Light on Lipids and Lipid-Modulating Enzymes in the Male Germline

**DOI:** 10.3390/jcm9020327

**Published:** 2020-01-23

**Authors:** Jessica L.H. Walters, Bart M. Gadella, Jessie M. Sutherland, Brett Nixon, Elizabeth G. Bromfield

**Affiliations:** 1Priority Research Centre for Reproductive Science, Schools of Environmental and Life Sciences and Biomedical Sciences and Pharmacy, Discipline of Biological Sciences, University of Newcastle, Callaghan, NSW 2308, Australia; 2Department of Biochemistry and Cell Biology, Faculty of Veterinary Medicine, Utrecht University, 3584 CM Utrecht, The Netherlands; 3Hunter Medical Research Institute, Pregnancy and Reproduction Program, New Lambton Heights, NSW 2305, Australia

**Keywords:** male infertility, lipids, lipid peroxidation, lipidome, oxidative stress, lipoxygenases, reactive oxygen species, spermatozoa

## Abstract

Despite the prevalence of male factor infertility, most cases are defined as idiopathic, thus limiting treatment options and driving increased rates of recourse to assisted reproductive technologies (ARTs). Regrettably, our current armory of ARTs does not constitute therapeutic treatments for male infertility, thus highlighting an urgent need for novel intervention strategies. In our attempts to fill this void, we have come to appreciate that the production of pathological levels of oxygen radicals within the male germline are a defining etiology of many idiopathic infertility cases. Indeed, an imbalance of reactive oxygen species can precipitate a cascade of deleterious sequelae, beginning with the peroxidation of membrane lipids and culminating in cellular dysfunction and death. Here, we shine light on the importance of lipid homeostasis, and the impact of lipid stress in the demise of the male germ cell. We also seek to highlight the utility of emerging lipidomic technologies to enhance our understanding of the diverse roles that lipids play in sperm function, and to identify biomarkers capable of tracking infertility in patient cohorts. Such information should improve our fundamental understanding of the mechanistic causes of male infertility and find application in the development of efficacious treatment options.

## 1. Introduction

Infertility is defined by the World Health Organization as the inability for a couple to conceive naturally following 12 months of unprotected intercourse. While research is now increasingly targeted towards improving our mechanistic understanding of the causes of infertility, a traditional lack of emphasis on the contribution of men to conception, embryo health, and early childhood development has left the field of male reproductive biology trailing behind that of the equivalent female field. Nevertheless, what we have come to appreciate is that male infertility is associated with numerous factors, including environmental and occupational exposures [1]; genetic mutations [2]; and diseases, such as testicular cancer [3]; and obesity [4]. Despite this knowledge, the vast majority (>70%) of male infertility cases are deemed idiopathic [5], a situation that severely limits therapeutic treatment strategies to rescue fertility. Over the last two decades, there has been a series of eminent publications focusing on the impact of oxidative stress on sperm cell biology and male fertility more broadly [6,7,8,9,10,11,12]. The causative links established in this body of work have in turn fueled growing interest in the use of antioxidant supplementation to improve sperm function and alleviate the burden of infertility (as reviewed by [13,14,15]). However, despite their therapeutic promise, trials of antioxidant interventions have been plagued by confounders, such as considerable variation in the implemented treatment regimens and a lack of targeted patient cohort selection, meaning that these strategies have been met with limited success (as reviewed [8]). These issues highlight the need for more stringent examination of both the underlying science and of the ensuing clinical trials. In this context, one of the most consistent intrinsic factors that contributes to cellular oxidative stress is membrane lipid oxidation.

While it is well known that extensive membrane lipid remodeling underpins the combined processes of germ cell development and their post-testicular maturation [16], it is somewhat surprising that the deleterious role of lipid stress has received minimal attention in terms of its contribution to sperm cell dysfunction. Indeed, despite our knowledge of the lipid composition of the sperm plasma membrane in many species [17], the contribution of these lipids to cellular stress and signal transduction has not been well studied in male germ cells. In contrast, the somatic cell literature is replete with examples of lipids fulfilling pivotal roles in signal transduction [18,19], inflammation [20], and cell death and disease [21]. By way of illustration, research into neurodegenerative diseases, such as Parkinson’s and Alzheimer’s disease, has begun to reveal lipidomic changes coinciding with disease onset [22,23,24]. Specifically, studies have reported that dietary intake of omega-3 poly-unsaturated fatty acids (PUFAs) may be associated with reduced Parkinson’s disease risk [25] while the lipid peroxidation product 4-hydroxynonenal (4HNE) has been reported in higher quantities in the cerebrospinal fluid of Parkinson’s patients [22,26]. Furthermore, recent lipidomic analyses have highlighted the presence of potential biomarkers for disease onset in blood samples between healthy controls and Alzheimer’s patients [27,28,29]. Similarly, lipids have increasingly been implicated in the pathophysiology of obesity [30,31], cardiovascular disease [32,33,34], diabetes [35], and several forms of cancer [36,37]. Accordingly, this knowledge is now being exploited to develop lipid-targeted therapies and membrane fortification strategies to protect cells against oxidative attack. Given this myriad of interest in lipid research, this narrative review serves to highlight important new roles for lipids in established models of disease, discuss the dynamic contribution of lipids to male germ cell development and function, and propose novel strategies to exploit analytical lipid technologies, such as redox lipidomics, to enhance our mechanistic understanding of male infertility.

## 2. Physiological and Pathophysiological Roles of Lipids

Lipids are amphiphilic molecules that fulfil a diversity of roles within the body. They are essential structural components of cell plasma (and organelle) membranes and in this capacity serve as key regulators of cellular homeostasis [38]. Among the most abundant and important classes of cell membrane lipids are the polyunsaturated fatty acids (PUFAs) of arachidonic (20:4), linoleic (18:2), and docosahexanoic acid (22:6), and the saturated fatty acids of myristic acid (14:0) and palmitic acid (16:0) [39]. Together with a glycerol backbone and a phosphate head group, these fatty acids are assembled into acyl tails to form phospholipids, such as phosphatidylethanolamine, phosphatidylcholine, and phosphatidylserine [40]. Phospholipids are the predominant lipid entity that delineate the plasma membrane, although this cellular barrier is also supplemented with numerous other structurally important lipids. These include sphingolipids, which are commonly found in regions of the outer membrane [40] and are involved in signaling [41], and sterols, such as cholesterol, which restrict membrane fluidity and impart structural support. Together, membrane lipids hold fundamental roles in signal transduction, membrane and organelle protection, and molecular trafficking in and out of cells [42,43]. It follows that disruption to membrane lipids, and the homeostatic influence they exert, can have profound downstream effects on human health and disease.

In numerous research fields, but particularly in cancer biology, the impact of cellular stress on lipid membranes and the resulting consequences for cell function has become a key focus for understanding cell death and disease [44]. Lipid peroxidation is a process that commonly occurs following the production of high levels of reactive oxygen species (ROS). ROS can activate phospholipase proteins, leading to the cleavage and liberation of PUFAs from membrane phospholipids [45,46]. The free PUFAs can, in turn, be broken down via a combination of non-enzymatic Fenton reactions [47], autoxidation [48], or enzymatic metabolism involving the action of lipoxygenases and/or cyclooxygenases [49]. Importantly, in many cell types, these combined catabolic pathways lead to a recently characterized cell death modality termed ferroptosis, so named on the basis of its iron dependency [21]. The ferroptotic pathway is initiated by the inactivation of the glutathione peroxidase 4 (GPX4), an antioxidant enzyme that affords membrane protection via the active reduction of lipid hydroperoxides [50]. It follows that GPX4 inactivation and/or depletion of its substrate glutathione (GSH) enables the accumulation of lipid hydroperoxides, the production of which is catalyzed by enzymes, such as lipoxygenases [51]. Recently, an alternative defense system based on the activity of ferroptosis-suppressor-protein 1 (FSP1) has been reported, which offers additional protection against lipid peroxidation and the ferroptosis cascade, even after GPX4 ablation [52]. Specifically, ubiquinone (CoQ10) targets and suppresses lipid peroxidation while FSP1, in conjunction with nitrate reductase (NAD(P)H), is responsible for replenishment of CoQ10 [52]. Ultimately, however, elevated levels of lipid hydroperoxides and their highly reactive metabolites (the lipid aldehydes) overwhelm the cellular defenses and result in ferroptosis, a form of caspase-independent cell death characterized by a unique gene expression profile, decreased mitochondrial size, and outer mitochondrial membrane rupture [21] (Figure 1).

Notably, ferroptosis has now been described as a characteristic feature of many diseases, including neurodegenerative disorders [53,54,55], ischemia [56,57], stroke [58,59], and numerous cancers [60,61]. Indeed, the extreme sensitivity of neurons to ferroptosis has been demonstrated by the targeted elimination of GPX4 in a mouse model, a strategy that gave rise to both neurodegeneration and paralysis [62]. Accordingly, pharmacological agents capable of modulating the activity of the ferroptosis pathway are gaining significant interest as potential targets to prevent disease progression [63]. Illustrative of this promise, recent studies have shown that the direct inhibition of ferroptosis can protect against cardiac injury [34,64]. Conversely, the sensitization of cancerous cells via the suppression of intrinsic ferroptotic inhibitors is proving an effective strategy to drive these cells toward a ferroptotic demise [65]. In one such study, pharmacological inhibition of nuclear factor erythroid 2-related factor 2 (NRF2; a protein that affords protection against ferroptosis owing to its role in GSH synthesis) effectively ameliorated the resistance of hepatocellular carcinoma cells to ferroptosis induced by either erastin or sorafenib [66]. Similarly, suppression of the miR-9, a microRNA implicated in the suppression of ferroptosis, significantly increased the potency of ferroptotic stimuli (i.e., RSL3 and erastin) in melanoma cells [67]. Thus, the ability to sensitize cancerous cells to programmed cell death through the induction of ferroptosis may provide an effective strategy to mitigate the risk posed by tumor growth and metastasis. Aside from their central role in the regulation of cellular death via ferroptosis, lipids also play key roles in alternative cell death modalities, such as those of necroptosis, pyroptosis, NETosis (as reviewed by [68,69]), and apoptosis (as reviewed [70]).

In addition to being implicated in multiple forms of cellular degeneration, it is well known that many diseases possess an altered lipid signature and one that may be unique to each condition. As reviewed by Long et al., altered lipid metabolism has been observed in a wide variety of cancers, including breast cancer, prostate cancer, leukemia, pancreatic cancer, and glioblastoma [37]. Furthermore, elevated levels of fatty acids, such as docosahexanoic acid, in blood plasma have been associated with a reduced risk of neurodegenerative disorders [71] whilst conversely, increased levels of sphingolipids have been reported in diabetic patients [35]. Similarly, changes in lipids docosahexanoic acid, eicosapentaenoic acid, docosapentanoic acid, and palmitoleic acid have been linked to fatty liver disease [72]. Notwithstanding these important observations, in many cases, a detailed understanding of the mechanisms underpinning lipid profile changes is lacking. However, for diseases, such as cardiovascular disease, where the knowledge of lipid homeostasis is more advanced, lipid biomarkers are now being utilized to predict the risk of atherosclerotic cardiovascular disease (ASCVD) [73]. Specifically, both observational and genetic evidence strongly support a causal relationship between high plasma concentrations of lipoprotein(a) and an increase in disease-related events, such as myocardial infarction, stroke, and valvular aortic stenosis [73,74,75]. Clinically, lipoprotein(a) levels of >100 nmol/L are considered indicative of an increased risk of ASCVD [73]. However, some discrepancies remain in the standardization of lipoprotein(a) assays and in the units used to report the levels of this lipid. Despite these challenges, the discovery of lipoprotein(a) as a predictive tool for ASCVD has led to the initiation of a randomized double-blind trial using antisense oligonucleotides to block the production of lipoprotein(a), as well as the development of other promising lipoprotein(a)-lowering therapies focused on small interfering RNA inhibitors [73]. This is just one example of the utility of lipid biology in informing novel diagnostics and interventions to prevent disease progression [36,76,77,78].

Among numerous other promising treatment strategies, one that is gaining substantial interest is the deuteration of fatty acids that reside within lipid membranes. The rationale for this approach rests with recent evidence that the sensitivity of fatty acids to lipid peroxidation reactions is primarily attributed to the presence of bis-allylic hydrogen sites, which are ideal targets for oxygen radicals to initiate lipid peroxidation [79]. Thus, the process of deuteration is achieved by the substitution of deuterium atoms in place of hydrogen atoms in the bis-allylic hydrogen sites of PUFAs (D-PUFAs), thus increasing the stability of these sites and mitigating the risk posed by lipid peroxidation. Importantly, this process has shown some promising proof-of-principle results in the prevention of diseases linked to neurodegenerative disorders and aging. Thus, the incorporation of D-PUFAs into the diet of *caenorhabditis elegans* (*c. elegans*) led to reductions in cellular stress and improvements in overall life span [79]. Furthermore, mouse studies have revealed memory improvements and decreased cellular stress following D-PUFA diet supplementation for models of Alzheimer’s and Huntington’s disease [80,81,82]. Thus, the treatment of diseases through the fortification of membranes within the cell is a promising approach as are the various strategies for lipid-based therapeutics summarized in Table 1. Importantly, this growth in understanding, technology, and appreciation for the role of lipids in health and disease has paved the way for a new chapter in fertility research targeted towards understanding the contribution of lipids to reproductive disorders, such as infertility [83,84]. Hereafter, we shall discuss our current understanding of lipid biology in male reproduction and highlight several areas for the continued growth of this field.

## 3. The Changing Profile of Lipids during Sperm Maturation

Spermatozoa are highly specialized cells that are formed in the testes through a process known as spermatogenesis [97]. During spermatogenesis, spermatogonial stem cells undergo multiple phases of mitotic and meiotic divisions before entering a complex remodeling process known as spermiogenesis. Collectively, the processes culminate in the production of morphologically mature spermatozoa, with a head domain containing the paternal genome and a flagellum responsible for the propagation of motility (Figure 2). Following testicular development, spermatozoa enter the male reproductive tract (epididymis) and begin an important phase of post-testicular maturation, during which they acquire the capacity for forward progressive motility [98] and shed their cytoplasmic droplets [99] before being stored in the distal epididymis in preparation for ejaculation [100]. Critically, a final stage of sperm maturation, termed capacitation, occurs in the female reproductive tract and is responsible for endowing the sperm cell with the ability to fertilize the ovulated oocyte [101,102]. Studies completed in numerous animal models have highlighted marked changes in the lipid composition of male germ cells during all stages of their development.

Key among these changes are alterations in the PUFA content as germ cells complete meiosis [103,104]. Specifically, the transformation of germ cells during their transition from spermatocytes to spermatids (Figure 2) is accompanied by a significant enrichment in the PUFA docosapentanoic acid [104]. This change occurs in concert with increases in the abundance of several alternative long chain PUFAs among the lipid content of round spermatids (Oresti et al., 2010). Whilst the physiological consequences of these changes have yet to be fully elucidated, their importance is alluded to by studies in mice, which have shown that a complete knockout of the delta-6 desaturase enzyme responsible for PUFA synthesis results in an infertility phenotype associated with spermatogenic failure [105]. Notably, however, supplementation with DHA, but not AA, to the diet of the mice was effective in rescuing this phenotype, leading to significant improvements in sperm concentration and morphology [105]. Similarly, male mice deficient in leptin receptors, key regulators of lipolysis, suffer from infertility and dysregulated spermatogenesis [106,107]. Moreover, dietary supplementation with medium chain triglycerides improved the fidelity of spermatogenesis, such that these animals showed improvements in epididymal sperm concentration and motility compared to their leptin deficient counterparts fed a control diet [106].

On their release from the testes, spermatozoa have been rendered both transcriptionally and translationally silent [108], yet still require substantial additional remodeling before gaining the functional competence to engage in oocyte interactions. It has long been known that this functional transformation is accompanied by pronounced changes in the lipid architecture of the cell [109,110], suggesting that dynamic lipid remodeling is an important facet of both epididymal maturation and capacitation. In early studies of the ram, it was demonstrated that epididymal spermatozoa possess a significant enrichment in ω−3 fatty acids, such as DHA, compared to that of their testicular counterparts while the opposite trend was observed for AA, the levels of which were instead significantly reduced during post-testicular maturation [109]. More recent work has concluded that the overall fatty acid content is increased in canine spermatozoa during their passage from the proximal (caput) to distal (cauda) segments of the epididymis [111]. The nature of this increase included enrichment of saturated fatty acids, mono-, and poly-unsaturated fatty acids (e.g., DHA) [111].

While the precise mechanisms responsible for promoting changes in the sperm lipid composition remain to be established, mounting interest has focused on the potential involvement of extracellular lipid vesicles or ‘epididymosomes’ [112], which are capable of delivering alternative cargo (e.g., proteins and small non-coding RNAs (sRNA)) to epididymal spermatozoa [113,114,115]. These extracellular vesicles possess high levels of cholesterol and sphingomyelin, which promote the formation of ordered membrane subdomains known as lipid rafts [112] and may play a role in coordinating their interaction with compatible sperm membrane domains [115]. While it is well known that extracellular vesicles are often enriched in lipids that differ from those of their parent cells [116], little is currently known regarding the lipid content of either the parent epididymal epithelial cells from which they originate or the epididymosomes themselves. Despite this, it is intriguing that the phospholipid content of epididymosomes differs based on the epididymal segment from which they are secreted [117]. Indeed, mouse epididymosomes isolated from the cauda epididymis are characterized by significantly lower proportions of phospholipids (such as phosphatidylcholine and phosphatidylethanolamine) but higher sphingomyelin than equivalent epididymosomes collected from the upstream caput segment [117]. Such changes coincide with alterations in the cholesterol to phospholipid ratio of epididymosomes [117], which mirror those recorded in epididymal spermatozoa. These findings encourage speculation that epididymosomes may regulate the lipid composition of epididymal spermatozoa in preparation for their extended storage in the male reproductive tract and their encounter with the female reproductive tract after ejaculation [113].

It is well established that the sperm ascent of the female reproductive tract is accompanied by a further wave of dynamic changes in their membrane lipid composition. Chief among these changes are the efflux of cholesterol and resultant increase in membrane fluidity, permeability, and fusibility characteristics, which signal the onset of capacitation [118,119,120]. Cholesterol removal is also permissive of membrane remodeling, including the repositioning of receptors and fusion machinery needed to prime the sperm cell for acrosomal exocytosis and downstream oocyte interactions [121,122,123,124]. Although the mechanisms by which sterols are depleted during capacitation is not established for all species, in porcine and mouse spermatozoa, bicarbonate-induced ROS formation appears to promote the oxidation of sterols at the sperm surface. The increased hydrophilicity of the oxysterol products so formed enhances their transfer to albumin acceptors [124]. Additionally, studies of human spermatozoa have reported the oxysterol, 25-hydroxycholesterol, as a potential biomarker of sperm function [125]. Indeed, in a lipidomic analysis of oxysterols, 25-hydroxycholesterol was found in the highest concentrations in normozoospermic sperm. Furthermore, 25-hydroxycholesterol levels positively correlated with sperm concentration [125].

Another consequence of cholesterol depletion from capacitating spermatozoa is the redistribution of lipid raft microdomains [126]. This redistribution appears to follow an anterior gradient such that lipid rafts, and their encapsulated cargo, tend to accumulate in the sperm head following the induction of capacitation [127,128]. It has been argued that this phenomenon positions sperm receptors appropriately for their interaction with cognate oocyte ligands during fertilization [129]. It follows that the tracking of key components of raft microdomains, such as the G_M1_ ganglioside, can provide important insight into the capacitation status of spermatozoa and potentially distinguish between fertile and infertile samples [130]. Furthermore, the application of high-resolution atomic force microscopy has allowed for visualization and tracking of key lipid components, such as those elements associated with membrane rafts, on the sperm surface during key stages of their functional maturation [131]. These collective studies demonstrate that lipids play an essential role in the development and maturation of the male gamete and accordingly, we shall next discuss established links between lipids and male fertility and review literature pertaining to the role of lipid-modulating enzymes in effecting changes in the sperm lipidome.

## 4. The Role of Lipids and Lipid-Associated Proteins in Spermatozoa and Infertility

Alongside the changing profile of lipids during sperm maturation, numerous studies have begun to highlight the important impact of lipid and lipid-modulating enzymes on fertility. As previously mentioned, lipids play an essential role in the cellular stress pathway that culminates in membrane breakdown and the production of highly reactive and cytotoxic lipid peroxidation products, such as aldehydes. Further, oxidative stress has long been established as a contributing factor to male infertility issues [8,132,133,134,135,136]. Recently, links have been drawn between the action of lipoxygenase enzymes and lipid peroxidation cascades within the male germline. Moreover, the targeted inhibition of arachidonate 15-lipoxygenase (ALOX15), with PD146176, has proven successful in reducing lipid peroxidation and cellular stress in both human and mouse germ cell models [137,138]. Additionally, PD146176 treatment can afford protection to human sperm functionality under conditions of oxidative stress, with notable improvements having been recorded in sperm motility, acrosome reaction rates, and adherence to the zonae pellucidae post-treatment [138].

Alternatively, round spermatids have been found to display acute sensitivity to ferroptosis induced by either erastin treatment or RSL3-mediated inhibition of GPX4 activity [139]. Importantly, this study also highlighted that the targeted inhibition of ACSL4 and ALOX15 (with rosiglitazone and PD146176, respectively) successfully protected round spermatids against lipid peroxidation and ferroptotic cell death [139]. Lipoxygenases have also been linked to infertility pathologies, such as asthenozoospermia (defined as low levels of sperm motility) [140,141]. Interestingly, levels of arachidonic acid, a dominant lipoxygenase substrate, have been reported as being 1.2-fold higher in asthenozoospermic spermatozoa compared to levels recorded in the sperm of healthy individuals [142]. Furthermore, the increased arachidonic acid in these samples was accompanied by an attendant l.5-fold increase in the ALOX15 metabolite, 15-HETE (15-hydroxyeicosatetraenoic acid), thus alluding to an important role for lipoxygenase-catalyzed metabolism of arachidonic acid within infertile patient samples [142].

Within the sperm cell itself, the distribution of PUFAs has been reported to vary between the head and the principle piece of the tail. Work completed on primate spermatozoa has uncovered dramatically increased levels of PUFAs in the sperm tail compared to the head, leading to the proposal that these lipids may modulate sperm motility via improved membrane fluidity [143]. In extrapolating this model, correlative links have been established between lipid profiles and sperm motility in porcine models, wherein PUFAs (docosahexanoic acid and docosapentanoic acid) were detected in significantly higher levels in spermatozoa with normal motility than those with poor motility [144]. Additionally, in human sperm cells, a large accumulation of PUFAs, such as DHA, was found to be present in the sperm head and is predicted to be involved in sperm maturation or interactions with the oocyte [145]. Moreover, patients presenting with idiopathic infertility had significantly lower levels of DHA as a proportion of total sperm lipids compared to high quality control cells pelleted by density gradient fractionation [145]. Notably, however, lower quality sperm partitioning within the density gradient were reported to have higher levels of both ω-3 and ω-6 fatty acids, irrespective of whether they originated from healthy or infertile donor samples [145]. This may be indicative of retention of the cytoplasmic droplet (found in immature spermatozoa) in these samples. Additionally, altered levels of stearic and polyunsaturated fatty acids within spermatozoa and seminal plasma samples have been reported in infertile patients [146]. Although additional work is clearly needed to establish reference values, this collective evidence suggests the utility of assessing lipid profiles as a potential strategy by which to screen the quality of an individual’s spermatozoa.

An additional dividend of this strategy is that altered lipid profiles associated with sperm dysfunction and infertility are likely influenced by the overall health status of an individual. By way of example, a study of patients suffering from the vision impairment, retinitis pigmentosa, were found to exhibit lower levels of DHA within their erythrocytes [147]. Furthermore, these patients also presented with significant reductions in DHA content in their spermatozoa and concomitant abnormal semen parameters, such as lowered sperm count and motility [147]. Other studies have drawn intriguing links between the interplay of environmental factors, lipid stress, and male fertility. Thus, a study focusing on patients with varicocele-induced infertility found the severity of this condition was exacerbated by exposure to cigarette smoke [148]. Specifically, the burden of DNA damage and lipid peroxidation was found to be increased in the spermatozoa of varicocele patients that smoked at moderate to heavy levels [148]. In a similar context, epidemiological studies have raised the prospect of a causative link between the lipid composition of human spermatozoa, overall semen quality, and a male’s body mass index (BMI) [149]. Thus, increased levels of sperm DHA were found to be positively correlated with normal sperm morphology while conversely, negative correlations were identified between sperm DHA and various sperm defects (including DNA damage) and between sperm DHA and BMI [149]. Such analyses are entirely consistent with an extensive body of literature highlighting the negative impacts of obesity on male fertility [150,151,152]. Furthermore, they also accord with data that dietary supplementation with different lipid formulations can influence sperm quality in both animal and human studies [153,154]. Such compelling evidence emphasizes the value of understanding body-wide lipid homeostasis in order to provide new insight into the dysfunction of sperm development and maturation that gives rise to idiopathic infertility. An important focus for such research may be the PUFA family, and in particular DHA, which are not only instrumental in the development of the male germline but are also commonly differentially accumulated in the dysfunctional gametes of male infertility patients (Table 2).

## 5. Analytical Lipid Technologies and Their Potential Application to Infertility Research

A common theme to emerge from our preceding summary of the contribution of lipids to male germ cell biology is that the application of lipid-based technologies to aid in the diagnosis, prevention, and understanding of male infertility is lagging far behind that of other heath disciplines. In this final section, we shall briefly outline how a resolute focus on lipids may enhance our understanding of idiopathic male infertility and reproductive health more broadly.

The structural and functional competence of biological membranes is achieved, in part, due to the astonishing diversification of phospholipids and their major components. Only through the advent of contemporary liquid chromatography mass spectrometry (LC-MS)-based lipidomics and the characterization of oxidatively modified lipids has this diversity been able to be appreciated. Glycerophospholipids are the major class of phospholipids, where one or two fatty acids are attached at the *sn*-1 (for saturated and mono-unsaturated fatty acids) or *sn*-2 positions (for PUFAs) of the glycerol backbone, with a polar group at the *sn*-3 position. Glycerophospholipids can then be further classified based on the nature of this polar group [155]. Most eukaryotic cells synthesize PUFAs from saturated fatty acids through the action of elongases, which add an ethylene group, or desaturases, which insert a double bond in the fatty acids (as reviewed by [155,156]). PUFAs have received considerable attention in reproductive biology due to the understanding that sperm cells become enriched in these easily oxidized substrates during maturation while simultaneously losing cytoplasmic antioxidant content during spermiogenesis. Here, it is the weak C-H bond at the *bis*-allylic position in PUFAs that is susceptible to hydrogen abstraction and forms the first intermediate of both enzymatic and non-enzymatic lipid peroxidation, the lipid radical [157]. Specific roles for oxygenated derivatives of PUFA, including leukotrienes and lipoxins, in the recruitment of immune cells and the resolution of inflammation have been able to be assigned through LC-MS protocols that use reverse phase LC and electrospray MS. Moreover, the oxidation of lipids also produces secondary products with shortened hydrocarbon chains (such as reactive electrophiles like 4HNE) that can be detected, despite their low abundance, using enrichment methods coupled to MS, such as biotin hydrazide affinity capture [158] and various click chemistry approaches [159]. While these approaches have not been widely used in reproductive biology, the burgeoning interest in lipid aldehydes and their ability to modify essential macromolecules in both the male [7,160,161] and female germline [162,163] will likely see the uptake of these technologies to answer questions surrounding germ cell aging and other reproductive pathologies.

While various types of MS have become the most accurate and sensitive quantitative methods for studies of lipid composition, the analysis of oxidized lipids remains a formidable task. The reasons for this are extensively discussed in excellent recent reviews by [155,164], but in short, this is due to the heterogeneity of the oxidized products, their susceptibility to degradation, the incredibly large number of isobaric oxidized lipid species, and until recently, a lack of accurate and available internal standards. The soft ionization techniques, electrospray ionization (ESI) and matrix-assisted laser desorption ionization (MALDI), have both been used extensively for the analysis of phospholipids and their oxidation products [165,166]. Coupled with LC and using a targeted approach to analysis and additional confirmation with fragmentation, ESI techniques have allowed for high specificity and sensitivity, though some oxidized isobaric species require further fragmentation or hydrolysis. Additionally, analysis of the hydrolyzed fatty acids may also be required to accurately determine the position of the oxygenated groups [167,168].

Often the level of structural detail required to map oxidation sites and the nature of isobaric oxidized lipid species is still refractory to high throughput or automated analysis. However, elegant two-dimensional chromatography approaches whereby lipids are separated first by class under the normal phase of hydrophilic interaction liquid chromatography (HILIC) conditions, and then further separated by their hydrophobicity by reverse-phase analysis in the second dimension has provided a powerful strategy to identify low-level oxidized lipid species [169]. Moreover, the development of higher resolution orbitrap instruments, such as the ThermoFisher Fusion Lumos, is highly permissive for the detection, unequivocal identification, and quantitation of oxidized phospholipids in cells, an example of which is described by [155].

Ultimately, the effectiveness and ease of uptake of these technologies relies on both the translation of the data collected into biologically relevant findings and the ability to integrate these data with those obtained of the lipid-modulating proteome, the metabolome, and the transcriptome. The complexity of redox modifications in the lipidome necessitates the need for more detailed systems biology approaches for lipid oxidation then those already well established for proteomics and transcriptomics. Bioinformatics packages, including LipidMatch [170] and LipidPioneer [171], for the interpretation of redox lipid data focus on pre-processing for peak alignment and integration, building an oxidized lipids database and automatically identifying them in LC-MS data. These tools (summarized in [164]), and importantly the development and updates of LipidMaps for the standardization of lipid analysis [168], have greatly aided data processing in many fields but still require some progress before they can be used effectively across all research fields. A great step forward has been attempts to integrate redox biology data into pathway analysis, with the Kyoto Encyclopedia of Genes and Genomes (KEGG) pathway now containing information on the role of oxidized lipids in lipid peroxidation, inflammation, linoleic and arachidonic acid metabolism, and ferroptosis, amongst others [155]. An example of the power of integrating lipidomic, proteomic, and transcriptomic data lies in a recent study by Parker et al., where lipid regulatory networks were examined in a large cohort of genetically distinct mouse strains to unveil new insight into the control and structure of mammalian lipid metabolism. This study established protein and genetic variants that are predicted to alter lipid abundance and has provided an important resource for probing lipid networks, especially in relation to hepatic lipotoxicity [172].

Despite the labor involved in ensuring accuracy and deriving meaning from complex lipidomic data, the potential advance in our understanding of reproductive biology and fertility warrants the growth of expertise necessary for the uptake of lipid technologies in our field. Furthermore, we are now in a unique position where the groundwork for these complex studies has largely been conducted in neighboring biological fields and permits the introduction of redox lipidomics and other lipid technologies to germ cells and reproductive tissues. Herein, we will deliberate on the potential use of these technologies to answer long-standing questions in reproduction (summarized in Figure 3).

## 6. Redox Lipidomics, Lipid-Targeted Antioxidants, and Male Infertility

The knowledge to be gained from performing a detailed study using redox lipidomics across developing germ cells and spermatozoa is immense considering the substantial proportion of infertility cases that display an oxidative mechanism. Moreover, the field of male reproductive health has been overwrought with antioxidant trials that are yet to yield substantial breakthroughs in therapeutics for infertile individuals [8]. While there are current efforts to manipulate enzymes, such as the lipoxygenases, to prevent widespread lipid peroxidation in germ cells and human spermatozoa [137,138], another attractive strategy is to use redox lipidomics to inform the development of targeted lipid-based antioxidants and/or to strategically fortify redox-sensitive sites via the deuteration of PUFAs. Interestingly, molecules that break the autoxidation of peroxyl radicals have proven capable of averting iron-dependent lipid peroxidation in other cell types. These radical trapping antioxidants (RTAs) inhibit phospholipid hydroperoxide formation and may hold promise for preventing membrane damage in germ cells. Although vitamin E is a natural RTA, a recent high-throughput screen has identified two novel RTAs, ferrostatin-1 and liproxstatin-1, which are comparatively more potent. These two RTAs have also proven highly effective within bilayer structures and are known to alleviate ferroptosis [173]. Additionally, synthetic compounds, such as nitroxide (Tempo), are also capable of inhibiting the production of hydroxyl radicals by blocking the Fenton reaction in mice [174] and warrant further analysis in male germ cells.

An important consideration in developing lipid-based antioxidants for infertility treatment is that the field is still lacking a comprehensive understanding of the lipid composition of germ cell and somatic cell membranes within the testis and how this changes both during spermatogenesis and throughout the course of life. While this is a difficult aim to achieve in humans due to limitations in our access to testis material, even mouse studies of the germ cell lipidome remain incomplete or entirely absent. Targeting both the germ cells and somatic cells of the testis will aid our understanding of the interconnectivity of germ cells and Sertoli cells during development and may also allow us to identify germ cell stage-specific markers of stress and how these change with the age of men or under redox stress conditions. Furthermore, applying lipidomics to a range of model species will assist in expanding our understanding of species-specific membrane composition to tailor ARTs, such as sperm cryopreservation or in vitro fertilisation (IVF), to species that do not respond well to these therapies.

Finally, the sensitivity of redox lipidomics technology is such that it has now been used to generate robust signatures of diverse cell death pathways, such as the oxidation of cardiolipins in apoptosis [175] and the presence of oxidized arachidonic and adrenic PE in ferroptosis [176]. This new knowledge provides essential leads to better understand cell death modalities in sperm cells and may allow us to rapidly characterize mechanisms that contribute to sperm cell death in infertile patients (Figure 3A). Moreover, an advanced understanding of cell death pathways under the control of lipid hydroperoxides may eventually be exploited to develop male-targeted contraceptives that are specific to meiotic or post-meiotic germ cells. While these goals will require extensive investigation and validation, the application of redox lipidomics has the potential to shine new light on many key issues of male reproductive health.

## 7. The Involvement of Lipids in the Biogenesis and Cargo Loading of Extracellular Vesicles

In the new literature surrounding somatic cell exosomes, it is appreciated that several lipid-related pathways are involved in the biogenesis of exosomes and contribute to the diverse contents of this class of extracellular vesicle. These topics are eloquently reviewed by [116,177,178]. However, it is notable that exosomes are enriched in desaturated molecular species of phospholipids, which account, in part, for their increased membrane rigidity compared to parent cell membranes. Accordingly, exosomes are known to be more resistant to detergent treatment than micro-vesicles, indicative of a higher membrane lipid order [177]. It has been known for some time that the disruption of plasma membrane lipid organization is critical to allow vesicle formation and the modification of the outer membrane leaflet by a cholesterol/sphingomyelin-binding protein promotes the formation of microvesicles. Interestingly, the translocation of phosphatidylserine [130] is also a prerequisite for the biogenesis of these microvesicles. New data regarding mechanisms known to enhance exosome production highlight the role of lipid transporters, such as ATP-binding cassette sub-family A member 3 (ABCA3), and the activity of the phospholipase D2 (PLD2), and diglyceride kinase. Conversely, the inhibition of phosphoinositide kinases, such as PI3 kinase, has a negative effect on exosome production. In focusing on the contribution of phospholipases, studies of the budding of micro-vesicles have given credence to the idea that the production of both exosomes and microvesicles could be coordinated by the phospholipases PLD1 and PLD2 [177,179,180]. In the organization of cargo into extracellular vesicles, raft-based microdomains appear important for the lateral segregation of cargo from within the endosomal membrane. Such microdomains are known to be enriched in sphingomyelinases from which ceramides can be formed through hydrolytic removal of the phosphocholine moiety. It is thought that the structure of ceramide may induce membrane curvature that in turn promotes domain-induced budding, implicating ceramide-dependent mechanisms in exosome biogenesis [177].

While there is a paucity of mechanistic knowledge regarding the biogenesis of exosomes from the male reproductive tract, the contribution of lipids to such biogenesis processes in reproductive cells is an untouched area of research. Despite this, several studies have contributed to an understanding of the lipid composition of human prostasomes and those of other species [181,182]. Moreover, intriguing effects of the paternal diet on the regulation of the sperm epigenome have been observed, highlighting clear alterations in the exosome cargo that are driven by diet [183]. What is missing from these important analyses is the impact of paternal diet on the lipid regulation of exosome biogenesis and cargo loading, where changes in the lipidome of the parent cells, their respective extracellular vesicles, and their vesicular lipid cargo are likely to provide a critical link between dietary perturbation and exosome content. Moreover, it will be critical to examine how dietary lipids may drive changes in exosome production, composition, and membrane fusion that are yet to be examined in the reproductive field (Figure 3B). This is an exciting area of research and one that has been made possible through the novel use of lipidomics and multiomics in the study of exosomes in many pioneering studies, including those of [182,184,185].

## 8. Mass Spectrometry Imaging of Lipids and Potential Applications for Reproductive Tissues

Having the tools to accurately detect oxidized phospholipids has led to a better understanding of their roles in both health and disease. However, it is not only the structure of these modified lipids but their concentration and tissue-specific localization that determine their function. Strikingly, this quest for a spatial understanding of lipids and metabolites within tissues has resulted in the rapid adoption of mass spectrometry imaging (MSI) techniques to lipidomics. MALDI-MSI is a well-established label-free technique that can be used to generate a highly specific, sensitive, and quantitative map of a broad range of biomolecules in cells and tissues [186,187]. The tissue used in these experiments is usually cryo-sectioned, mounted, and subsequently coated with a matrix to extract analytes from the tissue to form co-crystallization. MALDI MS is then used to scan specific regions of the tissue in an array of discrete points or ‘pixels’ and images are reconstructed to MS charts from these points [155]. These pixels can range from microns to nanometers depending on the instrument type, sample preparation, and analyte abundance [188]. MALDI-MSI has already been demonstrated to be a powerful technique for the spatial localization of phospholipids across many tissue types, and its range of clinical applications is expanding to fill an important gap between high throughput ‘-omics’ technologies and classic histology [189]. A very recent example of its use in a discovery/pre-clinical setting is the identification of lipid markers for traumatic brain injury, where the acylcarnitines (often indicative of mitochondrial damage) were revealed to be key markers and co-localized with microglia in the brain [190]. Importantly, these authors also identified that an increase in acylcarnitine lipids could be found in the region of the brain affected by Parkinson’s disease [190].

While protocols are still largely under development for the imaging of oxidized lipids, contemporary MALDI and gas cluster ion beam-secondary ion mass spectrometry (GCIB-SIMS) imaging may permit a high enough spatial resolution for this to be a distinct reality [155]. Additionally, these technologies are now being used to image lipids in single cells and to achieve the subcellular imaging of individual lipids. Indeed, recent work in single cells has captured the 3D spatial distribution of phospholipid classes, including PC, PE, and PI, in newly fertilized zebrafish across various stages of embryo development [191]. This was made possible through high spatial resolution MALDI protocols that can achieve a resolution of 5 μm [191]. In the context of the testis, it is easy to envisage many applications of MSI to understand the spatial distribution of lipids following invasive surgical procedures or to monitor lipid damage following torsion events or varicocele in men (Figure 3C). Moreover, single cell MSI and the monitoring of lipid peroxidation products in testis tissue would provide incredible insight into germ cell pathologies and could potentially provide a novel means to track molecules across the blood–testis barrier. MSI could also be used to monitor the effectiveness of new therapies that are targeted towards lipid stability or the prevention of lipid hydroperoxide production. While some of these experiments rely on new developments in technique sensitivity and the progress of oxidized lipid imaging, MSI has already been used for the spatial localization and quantitation of androgens in the mouse testis in a proof-of-concept experiment [192]. Similar to current applications of MSI for the localization of drugs across the blood–brain barrier, visualization of molecules and metabolites in the testis may become a new way to understand either the penetrability of the blood–testis barrier, or the ability of new drugs to reach their target sites. Finally, the use of lipid and metabolite MSI may provide an early indication of damage to reproductive tissues following the administration of novel cancer therapies to patients where there is currently no application to monitor membrane or lipid health following such procedures.

## 9. Conclusions

In summary, here, we have sought to highlight the diversity of important physiological roles that lipids fulfil in the maintenance of cellular homeostasis. These include fueling the bioenergetics of germ cell metabolism and the dynamic remodeling of germ cell architecture during their functional maturation. We also described the pathological consequences arising from dysregulation of lipid homeostasis and the prospect of utilizing lipid signatures as biomarkers of male factor infertility. In view of these roles, we propose that tangible benefits will flow from increased attention being devoted to the study of sperm lipid composition and the mechanisms responsible for promoting lipidomic changes in the spermatozoa of infertile patients. Indeed, driven by technological advances in lipid-based analytical tools, we are now presented with an exciting window of opportunity to refine our understanding of sperm cell biology. Such knowledge should equip us with rational strategies to diagnose and make progress towards preventing male factor infertility.

## Figures and Tables

**Figure 1 jcm-09-00327-f001:**
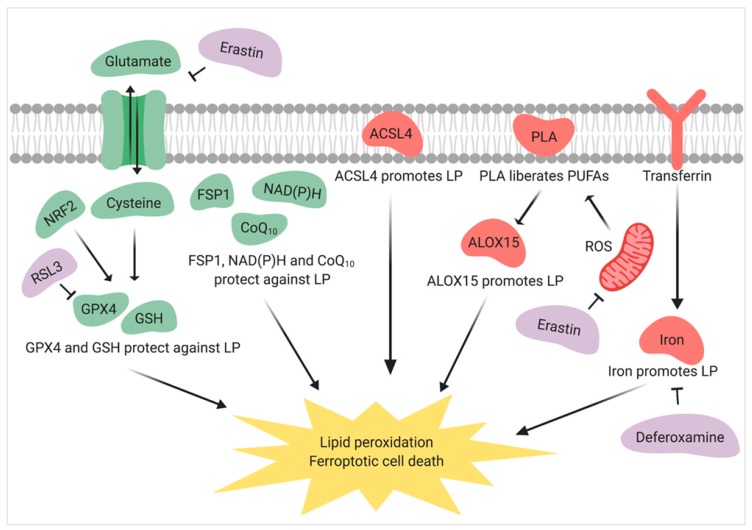
Key hallmarks of the ferroptotic pathway of cell death. Green and red indicate protectors and promoters of ferroptosis, respectively. Purple indicates chemical modulators of the ferroptosis pathway. Lipid peroxidation (LP) commonly occurs following the excessive production of mitochondrial reactive oxygen species (ROS). ROS activate phospholipase (PLA) enzymes, which then assist in the cleavage of poly-unsaturated fatty acids (PUFAs) from membrane phospholipids. Liberated PUFAs are catabolized via the enzymatic action of lipoxygenase proteins, such as arachidonate 15-lipoxygenase (ALOX15), non-enzymatic Fenton reactions, or autoxidation. Acyl-coenzyme A (Acyl-CoA) synthetase long-chain family member 4 (ACSL4) assists in sensitizing cells to ferroptosis by virtue of its role in lipid biosynthesis. Together with lysophosphatidylcholine acyltransferase 3 (LPCAT3), these lipid remodeling enzymes can generate and incorporate, respectively, long chain PUFAs in cell membranes, the key substrates for peroxidation. Transferrin transports iron into the cell, which promotes lipid peroxidation, while System X_c_^−^ imports/exports cystine and glutamate amino acids in and out of the cell, respectively. Once within the cell, cystine is converted to cysteine and together with nuclear factor erythroid 2-related factor 2 (NRF2) supports glutathione (GSH) synthesis. Glutathione peroxidase 4 (GPX4) activity is regulated by GSH and offers a first line of protection against ferroptosis by virtue of its reduction of both ROS and lipid peroxides. Interactions between ferroptosis-suppressor-protein 1 (FSP1), ubiquinone (CoQ_10_), and nitrate reductase (NAD(P)H) provide additional protection against ferroptosis by reducing lipid peroxidation. However, in the event that lipid peroxidation exceeds that of the intrinsic cellular defenses, a ferroptotic cell death ensues. Ferroptosis can be experimentally induced by erastin, which interferes with mitochondrial function and accentuates ROS production, or by disrupting system X_c_^−^ function. Alternatively, ferroptosis can be promoted by RSL3, a selective GPX4 inhibitor. Ferroptosis can also be inhibited with deferoxamine which disrupts iron activity. Image created using BioRender.com.

**Figure 2 jcm-09-00327-f002:**
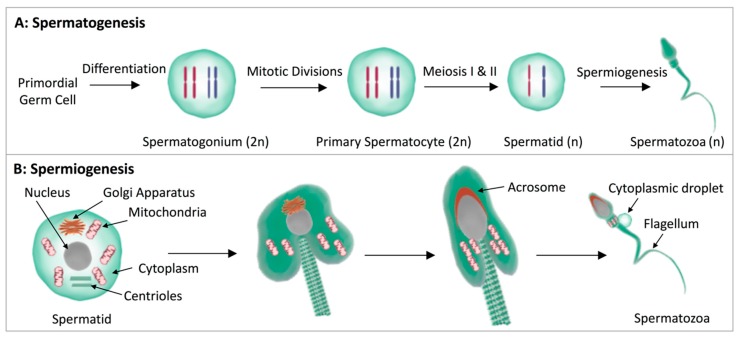
Testicular development of spermatozoa. Schematic illustration of (**A**) the successive phases of spermatogenesis (n: haploid, 2n: diploid, chromosomes are depicted in blue and red), and (**B**) spermiogenesis, which culminate in the formation of morphologically mature spermatozoa. During spermiogenesis, the Golgi apparatus gives rise to the acrosomal vesicle, the nucleus condenses, the centrioles form the flagellum, the mitochondria migrate to form the mitochondrial sheath, and excess cytoplasm is shed and accumulates to form a cytoplasmic droplet. Image created using symbols and adapted symbols from Luk and Idoya from Somersault18:24 Symbols (Creative commons license: CC BY-NC-SA 4.0).

**Figure 3 jcm-09-00327-f003:**
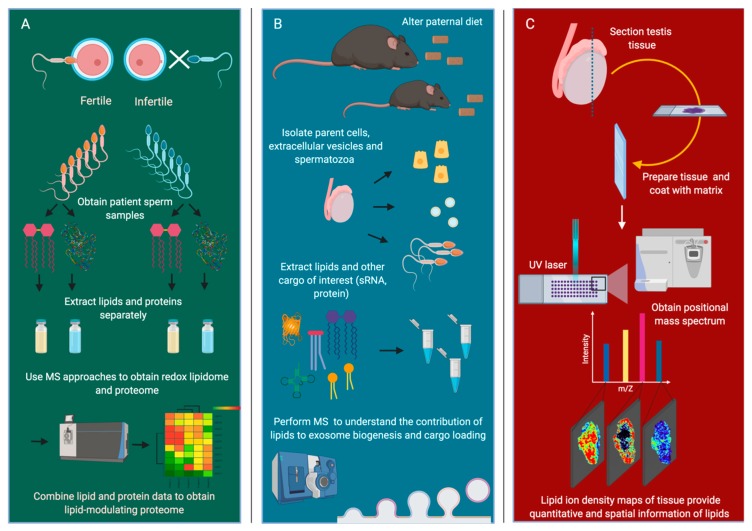
Applications for lipidomics in reproductive biology. (**A**) Multi-omics approaches are yet to be used to understand key differences between fertile and infertile sperm samples. Here, we propose the combination of redox lipidomics and proteomics will yield important insights into the lipid changes that form the basis of infertility and the enzymes that may be responsible for these changes. (**B**) While the influence of paternal diet on perturbations in small non-coding RNA (sRNA) cargo of epididymosomes has been the subject of several recent studies, the lipid cargo of these exosomes and of the parent cells following changes in paternal diet have not been examined. Lipidomics approaches may guide our understanding of exosome biogenesis and cargo loading into extracellular vesicles. (**C**) Mass spectrometry imaging has recently been coupled with lipidomics approaches, allowing a spatial understanding of quantitative lipid changes within tissue. This could be applied to testis tissue to understand region-specific lipid peroxidation or to track the localization of metabolites, hormones, and drugs across the blood–testis barrier to aid in the design of testis-targeted therapies. Image created with BioRender.com.

**Table 1 jcm-09-00327-t001:** Summary of current lipid-based strategies to combat disease.

Disease	Technology	Evidence ^1^
Neurological disorders	Manipulation of ferroptosis	• Deletion of GPX4 in an AD mouse model led to memory and learning deficits [54].
		• Significant improvements to locomotive activity in mice and decreases in ferroptotic cell death were observed following use of ferrostatin-1 in a mouse model for Parkinson’s disease [85].
	Deuteration	• Initial studies of *C. elegans* supplemented with D-PUFAs show reduced cellular stress as measured by ROS and lipid peroxidation levels. This treatment subsequently improved the lifespan, highlighting the promise for the prevention of age-related disorders [79].
		• A diet supplemented with D-PUFAs significantly improved memory performance in an AD mouse model [80].
		• A D-PUFA diet in a Huntington’s disease mouse model resulted in improvements to memory recognition and reduction in lipid peroxidation markers [81].
		• A mouse model for AD confirmed D-PUFA supplementation as a promising strategy to lower amyloid β-peptide production but did not improve learning deficits [82].
	Lipidomics and novel biomarkers	• An extensive lipidomic approach has identified 35 potential lipid biomarkers that varied between healthy controls and AD blood samples [27].
		• Blood lipidomics between aged healthy individuals and those with AD has identified 24 biomarkers that could be used to confirm AD with >70% accuracy [28].
		• The levels of six lipid peroxidation markers were monitored between healthy and AD blood samples to provide a promising model for AD diagnosis [29].
Cardiovascular and lung diseases	Manipulation of ferroptosis	• A COPD mouse model induced via cigarette exposure demonstrated that *GPX4* gene deletion resulted in an exacerbation of hallmark features of COPD and increased lipid peroxidation and ferroptotic cell death [86].
		• Using a radiation-induced lung fibrosis (RILF) mouse model, GPX4 levels were shown to be significantly reduced compared to healthy controls. Further, the addition of the ferroptosis inhibitor liproxstatin-1 lowered levels of cellular stress and improved the GPX4 concentration [87].
	Lipidomics and novel biomarkers	• A lipidomic study was completed on 1028 subjects to identify lipid metabolites indicative of risk for coronary heart disease. Metabolites identified included lysophosphatidylcholine 18:1, lysophosphatidylcholine 18:2, monoglyceride 18:2, and sphingomyelin 28:1 [88].
		• A study of 220 individuals highlighted unique differences in the lipid profiles between unstable and stable coronary heart disease [89].
		• A lipidomic study completed on 685 blood samples highlighted that the relative risk of cardiovascular disease was associated with increased levels of cholesterol esters and triacylglycerols [90].
		• The identification of lipoprotein(a) as a risk factor for ASCVD has led to a clinical trial set to begin in 2020, which will examine the possibility of targeting lipoprotein(a) production to protect against the disease [73].
Cancer	Lipidomics and novel biomarkers	• Screening of almost 20,000 individuals found that colorectal adenomas (advanced and non-advanced) were associated with increased levels of triglycerides while ApoA-1 and HDL cholesterol were linked to non-advanced adenomas [91].
		• A positive relationship has been observed between phosphatidylserine and lyso-phosphatidylserine and lung cancer prevalence and a negative correlation with lyso-phosphatidylethanolamine and phosphatidylethanolamine and lung cancer. Furthermore, this study identified that the lipidomic profile varied between different subtypes of lung cancer [92].
		• A lipidomic analysis identified 64 potential lipid biomarkers that were either up or downregulated in the presence of colorectal cancer [93].
		• A lipidomic analysis comparing prostate cancer patients with healthy controls identified 35 potential lipid biomarkers for diagnostic use [94].
	Manipulation of ferroptosis	• A recent study confirmed SKBr3 breast cancer cells as sensitive to ferroptosis using the ferroptosis inhibitors deferoxamine and ferrostatin-1 [95].
		• A study confirmed the sensitivity of acute lymphoblastic leukemia cells to ferroptosis induced through RSL3 treatment. Furthermore, ferroptosis and lipid peroxidation were prevented through Ferrostatin-1 treatment and lipoxygenase inhibition [96].

^1^ Abbreviations: arachidonate 15-lipoxygenase (ALOX15); acyl-CoA synthetase long-chain family member 4 (ACSL4); nuclear factor erythroid 2-related factor 2 (NRF2); chronic obstructive pulmonary disease (COPD); atherosclerotic cardiovascular disease (ASCVD); deuterium-reinforced polyunsaturated fatty acids (D-PUFAs), Alzheimer’s disease (AD); high-density lipoprotein (HDL); apolipoprotein A-1 (ApoA-1); radiation-induced lung fibrosis (RILF).

**Table 2 jcm-09-00327-t002:** Summary of key PUFAs represented within the male germline.

PUFA	Temporal Accumulation and Consequences of Dysregulation ^1^
Docosahexanoic acid (22:6)	↑ Increased levels of DHA in ram spermatozoa collected from the epididymis compared to the testes [109].
	↑ Increased levels of DHA in dog spermatozoa isolated from the distal versus proximal epididymis [111].
	↑ Extremely high levels of DHA found in the monkey sperm tail compared to the sperm head [143].
	↓ Reduced levels of DHA correlated with low motility in boar spermatozoa compared to normal motility controls [144].
	↓ Significantly lower levels of DHA present in patients with asthenozoospermia and oligozoospermia compared to normozoospermic controls [145].
	↓ Significantly lower levels of DHA reported in infertile human semen samples compared to healthy controls [146].
	↓ Patients presenting with retinitis pigmentosa had significantly lower levels of DHA within their sperm [147].
	↓ A negative correlation was reported between human sperm DHA levels, DNA damage, and BMI; DHA content was correlated positively with normal semen parameters such as sperm count, vitality and motility [149].
Docosapentanoic acid (22:5)	↑ Increased levels of DPA accompanies transition of rat germ cells from spermatocytes to post-meiotic spermatids [104].
	↓ Reduced levels of DPA correlated with boar spermatozoa presenting with low motility compared to normal motility controls [144].
Eicosapentaenoic acid (20:5)	↓ Significantly lower levels reported in human semen samples from infertile individuals compared to healthy controls [146]
Arachidonic acid (20:4)	↑ Increased levels of AA were observed in the seminal plasma of human patient samples with asthenozoospermia compared to healthy controls [142].
	↑ Increased levels of AA found in the tails compared to the heads of monkey spermatozoa [143].
	↓ Reduced AA levels were observed in ram spermatozoa from the epididymis compared to the testes [109].
	↓ Patients presenting with retinitis pigmentosa had significantly lower levels of AA within their sperm [147].
Dihomo-γ-linolenic acid (20:3)	↑ Higher levels of DGLA found in the tails compared to the heads of monkey spermatozoa [143].
	↑ Significantly higher levels of DGLA were reported in infertile human semen samples compared to healthy controls [146].
	↓ Patients presenting with retinitis pigmentosa had significantly lower levels of DGLA within their spermatozoa [147].
γ-Linolenic acid (18:3)	↑ Higher levels of LA found in the tails compared to the heads of monkey spermatozoa [143].
	↑ Significantly higher levels of LA present in the spermatozoa of patients suffering from asthenozoospermia and oligozoospermia compared to normozoospermic controls [145].

^1^↑↓ Indicates an increase or decrease in PUFA content, respectively. Abbreviations: docosahexanoic acid (DHA); docosapentanoic acid (DPA); arachidonic acid (AA); lineiolic acid (LA); dihomo-γ-linolenic acid (DGLA).

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
