# Peer review of "Male Infertility: Shining a Light on Lipids and Lipid-Modulating Enzymes in the Male Germline"

_jcm, 2020, doi:10.3390/jcm9020327_

Round 1

Reviewer 1 Report

Interresting expert (narrative) review that formulates many hypotheses.

The major point is that it is not a systematic review.

Altough it seems to be exhaustive, we cannot conclude that it is the state of the art. Finally the synthesis is unclear.

Some little problems in the typo:

p2 l47, add a parenthesis after [13-15]

p2 l50, as and not As

p2 l67 highlighted

p2 l72, narrative review

p4 l161, a wide

Reviewer 2 Report

Numerous current studies in the area of human reproduction and especially the problem of decreased male fertility focus on the investigation of the genome and proteome. As the authors rightly point out, the field of lipidomics is fairly neglected. The article highlights the importance of the elucidation of lipid profiles of spermatozoa membranes and exosomes, summarizing the currently available data and perspectives of further studies. The manuscript is well presented and legible, will be valuable reading for both researchers and clinicians.

I have found few typesetting errors:

l.159: it is well known…

Table 1: the line separating neurological and cardiovascular diseases should be extended to the first column

390: for saturated and mono-unsaturated…

Fig. 2: Skip # at A; and B: descriptions

Reviewer 3 Report

The review by Walters and coworkers provides an exhaustive analysis on the effect of lipids on the male germline, establishing a close connection between lipid modulating enzymes and the idiopathic male infertility. Authors also highlight the utility of lipidomic technologies to clarify the diverse roles that lipids play in sperm functionality, to identify biomarkers capable of tracking male infertility and, finally, to find application in the development of successful treatment options. The review is interesting, well written. In line 539 of the manuscript, the reference should be written as a number (maybe 177?)